# Using multiple samples to learn mixture models

**Jason Lee**[*]
Stanford University
Stanford, USA
jdl17@stanford.edu

**Ran Gilad-Bachrach**
Microsoft Research
Redmond, USA
rang@microsoft.com

**Rich Caruana**
Microsoft Research
Redmond, USA
rcaruana@microsoft.com

## Abstract

In the mixture models problem it is assumed that there are $K$ distributions $\theta_1, \ldots, \theta_K$ and one gets to observe a sample from a mixture of these distributions with unknown coefficients. The goal is to associate instances with their generating distributions, or to identify the parameters of the hidden distributions. In this work we make the assumption that we have access to several samples drawn from the same $K$ underlying distributions, but with different mixing weights. As with topic modeling, having multiple samples is often a reasonable assumption. Instead of pooling the data into one sample, we prove that it is possible to use the differences between the samples to better recover the underlying structure. We present algorithms that recover the underlying structure under milder assumptions than the current state of art when either the dimensionality or the separation is high. The methods, when applied to topic modeling, allow generalization to words not present in the training data.

## 1 Introduction

The mixture model has been studied extensively from several directions. In one setting it is assumed that there is a single sample, that is a single collection of instances, from which one has to recover the hidden information. A line of studies on clustering theory, starting from [5] has proposed to address this problem by finding a projection to a low dimensional space and solving the problem in this space. The goal of this projection is to reduce the dimension while preserving the distances, as much as possible, between the means of the underlying distributions. We will refer to this line as MM (Mixture Models). On the other end of the spectrum, Topic modeling (TM), [9, 3], assumes multiple samples (documents) that are mixtures, with different weights of the underlying distributions (topics) over words.

Comparing the two lines presented above shows some similarities and some differences. Both models assume the same generative structure: a point (word) is generated by first choosing the distribution $\theta_i$ using the mixing weights and then selecting a point (word) according to this distribution. The goal of both models is to recover information about the generative model (see [10] for more on that). However, there are some key differences:

(a) In MM, there exists a single sample to learn from. In TM, each document is a mixture of the topics, but with different mixture weights.

(b) In MM, the points are represented as feature vectors while in TM the data is represented as a word-document co-occurrence matrix. As a consequence, the model generated by TM cannot assign words that did not previously appear in any document to topics.

---

[*]Work done while the author was an intern at Microsoft Resaerch

(c) TM assumes high density of the samples, i.e., that the each word appears multiple times. However, if the topics were not discrete distributions, as is mostly the case in MM, each "word" (i.e., value) would typically appear either zero or one time, which makes the co-occurrence matrix useless.

In this work we try to close the gap between MM and TM. Similar to TM, we assume that multiple samples are available. However, we assume that points (words) are presented as feature vectors and the hidden distributions may be continuous. This allows us to solve problems that are typically hard in the MM model with greater ease and generate models that generalize to points not in the training data which is something that TM cannot do.

## 1.1  Definitions and Notations

We assume a mixture model in which there are $K$ mixture components $\theta_1, \ldots, \theta_K$ defined over the space $X$. These mixture components are probability measures over $X$. We assume that there are $M$ mixture models (samples), each drawn with different mixture weights $\Phi^1, \ldots, \Phi^M$ such that $\Phi^j = (\phi_1^j, \ldots, \phi_K^j)$ where all the weights are non-negative and sum to 1. Therefore, we have $M$ different probability measures $D_1, \ldots, D_M$ defined over $X$ such that for a measurable set $A$ and $j = 1, \ldots, M$ we have $D_j(A) = \sum_i \phi_i^j \theta_i(A)$. We will denote by $\phi_{\min}$ the minimal value of $\phi_i^j$.

In the first part of this work, we will provide an algorithm that given samples $S_1, \ldots, S_M$ from the mixtures $D_1, \ldots, D_M$ finds a low-dimensional embedding that preserves the distances between the means of each mixture.

In the second part of this work we will assume that the mixture components have disjoint supports. Hence we will assume that $X = \cup_j C_j$ such that the $C_j$'s are disjoint and for every $j$, $\theta_j(C_j) = 1$. Given samples $S_1, \ldots, S_M$, we will provide an algorithm that finds the supports of the underlying distributions, and thus clusters the samples.

## 1.2  Examples

Before we dive further in the discussion of our methods and how they compare to prior art, we would like to point out that the model we assume is realistic in many cases. Consider the following example: assume that one would like to cluster medical records to identify sub-types of diseases (e.g., different types of heart disease). In the classical clustering setting (MM), one would take a sample of patients and try to divide them based on some similarity criteria into groups. However, in many cases, one has access to data from different hospitals in different geographical locations. The communities being served by the different hospitals may be different in socioeconomic status, demographics, genetic backgrounds, and exposure to climate and environmental hazards. Therefore, different disease sub-types are likely to appear in different ratios in the different hospital. However, if patients in two hospitals acquired the same sub-type of a disease, parts of their medical records will be similar.

Another example is object classification in images. Given an image, one may break it to patches, say of size 10x10 pixels. These patches may have different distributions based on the object in that part of the image. Therefore, patches from images taken at different locations will have different representation of the underlying distributions. Moreover, patches from the center of the frame are more likely to contain parts of faces than patches from the perimeter of the picture. At the same time, patches from the bottom of the picture are more likely to be of grass than patches from the top of the picture.

In the first part of this work we discuss the problem of identifying the mixture component from multiple samples when the means of the different components differ and variances are bounded. We focus on the problem of finding a low dimensional embedding of the data that preserves the distances between the means since the problem of finding the mixtures in a low dimensional space has already been address (see, for example [10]). Next, we address a different case in which we assume that the support of the hidden distributions is disjoint. We show that in this case we can identify the supports of each distribution. Finally we demonstrate our approaches on toy problems. The proofs of the theorems and lemmas

appear in the appendix. Table 1 summarizes the applicability of the algorithms presented here to the different scenarios.

## 1.3 Comparison to prior art

There are two common approaches in the theoretical study of the MM model. The method of moments [6, 8, 1] allows the recovery of the model but requires exponential running time and sample sizes. The other approach, to which we compare our results, uses a two stage approach. In the first stage, the data is projected to a low dimensional space and in the second stage the association of points to clusters is recovered. Most of the results with this approach assume that the mixture components are Gaussians. Dasgupta [5], in a seminal paper, presented the first result in this line.

|  | Disjoint clusters | Overlapping clusters |
|---|---|---|
| **High dimension** | DSC, MSP | MSP |
| **Low dimension** | DSC | |

Table 1: Summary of the scenarios the MSP (Multi Sample Projection) algorithm and the DSC (Double Sample Clustering) algorithm are designed to address.

He used random projections to project the points to a space of a lower dimension. This work assumes that separation is at least $\Omega(\sigma_{\max}\sqrt{n})$. This result has been improved in a series of papers. Arora and Kannan [10] presented algorithms for finding the mixture components which are, in most cases, polynomial in $n$ and $K$. Vempala and Wang [11] used PCA to reduce the required separation to $\Omega\left(\sigma_{\max}K^{1/4}\log^{1/4}\left(n/\phi_{\min}\right)\right)$. They use PCA to project on the first $K$ principal components, however, they require the Gaussians to be spherical. Kanan, Salmasian and Vempala [7] used similar spectral methods but were able to improve the results to require separation of only $c\sigma_{\max}K^{2/3}/\phi_{\min}^2$. Chaudhuri [4] have suggested using correlations and independence between features under the assumption that the means of the Gaussians differ on many features. They require separation of $\Omega\left(\sigma_{\max}\sqrt{K\log(K\sigma_{\max}\log n/\phi_{\min})}\right)$, however they assume that the Gaussians are axis aligned and that the distance between the centers of the Gaussians is spread across $\Omega\left(K\sigma_{\max}\log n/\phi_{\min}\right)$ coordinates.

We present a method to project the problem into a space of dimension $d^*$ which is the dimension of the affine space spanned by the means of the distributions. We can find this projection and maintain the distances between the means to within a factor of $1 - \epsilon$. The different factors, $\sigma_{\max}$, $n$ and $\epsilon$ will affect the sample size needed, but do not make the problem impossible. This can be used as a preprocessing step for any of the results discussed above. For example, combining with [5] yields an algorithm that requires a separation of only $\Omega\left(\sigma_{\max}\sqrt{d^*}\right) \leq \Omega\left(\sigma_{\max}\sqrt{K}\right)$. However, using [11] will result in separation requirement of $\Omega\left(\sigma_{\max}\sqrt{K\log\left(K\sigma_{\max}\log d^*/\phi_{\min}\right)}\right)$. There is also an improvement in terms of the value of $\sigma_{\max}$ since we need only to control the variance in the affine space spanned by the means of the Gaussians and do not need to restrict the variance in orthogonal directions, as long as it is finite. Later we also show that we can work in a more generic setting where the distributions are not restricted to be Gaussians as long as the supports of the distributions are disjoint. While the disjoint assumption may seem too strict, we note that the results presented above make very similar assumptions. For example, even if the required separation is $\sigma_{\max}K^{1/2}$ then if we look at the Voronoi tessellation around the centers of the Gaussians, each cell will contain at least $1 - (2\pi)^{-1}K^{3/4}\exp\left(-K/2\right)$ of the mass of the Gaussian. Therefore, when $K$ is large, the supports of the Gaussians are almost disjoint.

## 2 Projection for overlapping components

In this section we present a method to use multiple samples to project high dimensional mixtures to a low dimensional space while keeping the means of the mixture components

---

**Algorithm 1 Multi Sample Projection (MSP)**

---

**Inputs:**
Samples $S_1, \ldots, S_m$ from mixtures $D_1, \ldots, D_m$
**Outputs:**
Vectors $\bar{v}_1, \ldots, \bar{v}_{m-1}$ which span the projected space
**Algorithm:**

1. For $j = 1, \ldots, m$ let $\bar{E}_j$ be the mean of the sample $S_j$

2. For $j = 1, \ldots, m-1$ let $\bar{v}_j = \bar{E}_j - \bar{E}_{j+1}$

3. return $\bar{v}_1, \ldots, \bar{v}_{m-1}$

---

well separated. The main idea behind the Multi Sample Projection (MSP) algorithm is simple. Let $\mu_i$ be the mean of the $i$'th component $\theta_i$ and let $E_j$ be the mean of the $j$'th mixture $D_j$. From the nature of the mixture, $E_j$ is in the convex-hull of $\mu_1, \ldots, \mu_K$ and hence in the affine space spanned by them; this is demonstrated in Figure 1. Under mild assumptions, if we have sufficiently many mixtures, their means will span the affine space spanned by $\mu_1, \ldots, \mu_K$. Therefore, the MSP algorithm estimates the $E_j$'s and projects to the affine space they span. The reason for selecting this sub-space is that by projecting on this space we maintain the distance between the means while reducing the dimension to at most $K - 1$. The MSP algorithm is presented in Algorithm 1. In the following theorem we prove the main properties of the MSP algorithm. We will assume that $X = \mathbb{R}^n$, the first two moments of $\theta_j$ are finite, and $\sigma_{\max}^2$ denotes maximal variance of any of the components in any direction. The separation of the mixture components is $\min_{j \neq j'} \|\mu_j - \mu_{j'}\|$. Finally, we will denote by $d^*$ the dimension of the affine space spanned by the $\mu_j$'s. Hence, $d^* \leq K - 1$.

**Theorem 1.** *MSP Analysis*

*Let $E_j = E[D_j]$ and let $v_j = E_j - E_{j+1}$. Let $N_j = |S_j|$. The following holds for MSP:*

1. *The computational complexity of the MSP algorithm is $n \sum_{j=1}^{M} N_j + 2n(m-1)$ where $n$ is the original dimension of the problem.*

2. *For any $\epsilon > 0$, $\Pr\left[\sup_j \left\|E_j - \bar{E}_j\right\| > \epsilon\right] \leq \frac{n\sigma_{max}^2}{\epsilon^2} \sum_j \frac{1}{N_j}$ .*

3. *Let $\bar{\mu}_i$ be the projection of $\mu_i$ on the space spanned by $\bar{v}_1, \ldots, \bar{v}_{M-1}$ and assume that $\forall i, \ \mu_i \in span\{v_j\}$. Let $\alpha_j^i$ be such that $\mu_i = \sum_j \alpha_j^i v_j$ and let $A = \max_i \sum \left|\alpha_j^i\right|$ then with probability of at least $1 - \frac{n\sigma_{max}^2}{\epsilon^2} \sum_j \frac{1}{N_j}$*

$$\Pr\left[\max_{i,i'} \left| \|\mu_i - \mu_{i'}\| - \|\bar{\mu}_i - \bar{\mu}_{i'}\| \right| > \epsilon\right] \leq \frac{4n\sigma_{max}^2 A^2}{\epsilon^2} \sum_j \frac{1}{N_j} \ .$$

The MSP analysis theorem shows that with large enough samples, the projection will maintain the separation between the centers of the distributions. Moreover, since this is a projection, the variance in any direction cannot increase. The value of $A$ measures the complexity of the setting. If the mixing coefficients are very different in the different samples then $A$ will be small. However, if the mixing coefficients are very similar, a larger sample is required. Nevertheless, the size of the sample needed is polynomial in the parameters of the problem. It is also apparent that with large enough samples, a good projection will be found, even with

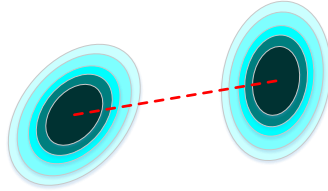

Figure 1: The mean of the mixture components will be in the convex hull of their means demonstrated here by the red line.

large variances, high dimensions and close centroids.

A nice property of the bounds presented here is that they assume only bounded first and second moments. Once a projection to a low dimensional space has been found, it is possible to find the clusters using approaches presented in section 1.3. However, the analysis of the MSP algorithm assumes that the means of $E_1, \ldots, E_M$ span the affine space spanned by $\mu_1, \ldots, \mu_K$. Clearly, this implies that we require that $m > d^*$. However, when $m$ is much larger than $d^*$, we might end-up with a projection on too large a space. This could easily be fixed since in this case, $\bar{E}_1, \ldots, \bar{E}_m$ will be almost co-planar in the sense that there will be an affine space of dimension $d^*$ that is very close to all these points and we can project onto this space.

# 3   Disjoint supports and the Double Sample Clustering (DSC) algorithm

In this section we discuss the case where the underlying distributions have disjoint supports. In this case, we do not make any assumption about the distributions. For example, we do not require finite moments. However, as in the mixture of Gaussians case some sort of separation between the distributions is needed, this is the role of the disjoint supports.

We will show that given two samples from mixtures with different mixture coefficients, it is possible to find the supports of the underlying distributions (clusters) by building a tree of classifiers such that each leaf represents a cluster. The tree is constructed in a greedy fashion. First we take the two samples, from the two distributions, and reweigh the examples such that the two samples will have the same cumulative weight. Next, we train a classifier to separate between the two samples. This classifier becomes the root of the tree. It also splits each of the samples into two sets. We take all the examples that the classifier assign to the label $+1(-1)$, reweigh them and train another classifier to separate between the two samples. We keep going in the same fashion until we can no longer find a classifier that splits the data significantly better than random.

To understand why this algorithm works it is easier to look first at the case where the mixture distributions are known. If $D_1$ and $D_2$ are known, we can define the $L_1$ distance between them as $L_1(D_1, D_2) = \sup_A |D_1(A) - D_2(A)|$.[1] It turns out that the supremum is attained by a set $A$ such that for any $i$, $\mu_i(A)$ is either zero or one. Therefore, any inner node in the tree splits the region without breaking clusters. This process proceeds until all the points associated with a leaf are from the same cluster in which case, no classifier can distinguish between the classes.

When working with samples, we have to tolerate some error and prevent overfitting. One way to see that is to look at the problem of approximating the $L_1$ distance between $D_1$ and $D_2$ using samples $S_1$ and $S_2$. One possible way to do that is to define $\hat{L}_1 = \sup_A \left| \frac{A \cap S_1}{|S_1|} - \frac{A \cap S_2}{|S_2|} \right|$.

However, this estimate is almost surely going to be 1 if the underlying distributions are absolutely continuous. Therefore, one has to restrict the class from which $A$ can be selected to a class of VC dimension small enough compared to the sizes of the samples. We claim that asymptotically, as the sizes of the samples increase, one can increase the complexity of the class until the clusters can be separated.

Before we proceed, we recall a result of [2] that shows the relation between classification and the $L_1$ distance. We will abuse the notation and treat $A$ both as a subset and as a classifier. If we mix $D_1$ and $D_2$ with equal weights then

$$
\begin{aligned}
\operatorname{err}(A) &= D_1(X \setminus A) + D_2(A) \\
&= 1 - D_1(A) + D_2(A) \\
&= 1 - (D_1(A) - D_2(A)) \ .
\end{aligned}
$$

Therefore, minimizing the error is equivalent to maximizing the $L_1$ distance.

**Algorithm 2** Double Sample Clustering (DSC)
---
**Inputs:**

- Samples $S_1, S_2$
- A binary learning algorithm L that given samples $S_1, S_2$ with weights $w_1, w_2$ finds a classifier $h$ and an estimator $e$ of the error of $h$.
- A threshold $\tau > 0$.

**Outputs:**

- A tree of classifiers

**Algorithm:**

1. Let $w_1 = 1$ and $w_2 = |S_1|/|S_2|$
2. Apply L to $S_1$ & $S_2$ with weights $w_1$ & $w_2$ to get the classifier $h$ and estimator $e$.
3. If $e \geq \frac{1}{2} - \tau$,
   (a) return a tree with a single leaf.
4. else
   (a) For $j = 1, 2$, let $S_j^+ = \{x \in S_j \text{ s.t. } h(x) > 0\}$
   (b) For $j = 1, 2$, let $S_j^- = \{x \in S_j \text{ s.t. } h(x) < 0\}$
   (c) Let $T^+$ be the tree returned by the DSC algorithm applied to $S_1^+$ and $S_2^+$
   (d) Let $T^-$ be the tree returned by the DSC algorithm applied to $S_1^-$ and $S_2^-$
   (e) return a tree in which $c$ is at the root node and $T^-$ is its left subtree and $T^+$ is its right subtree

---

The key observation for the DSC algorithm is that if $\phi_i^1 \neq \phi_i^2$, then a set $A$ that maximizes the $L_1$ distance between $D_1$ and $D_2$ is aligned with cluster boundaries (up to a measure zero).

Furthermore, $A$ contains all the clusters for which $\phi_i^1 > \phi_i^2$ and does not contain all the clusters for which $\phi_i^1 < \phi_i^2$. Hence, if we split the space to $A$ and $\bar{A}$ we have few clusters in each side. By applying the same trick recursively in each side we keep on bisecting the space according to cluster boundaries until subspaces that contain only a single cluster remain. These sub-spaces cannot be further separated and hence the algorithm will stop. Figure 2 demonstrates this idea. The following lemma states this argument mathematically:

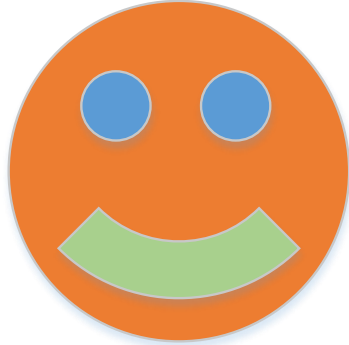

**Lemma 1.** *If $D_j = \sum_i \phi_i^j \theta_i$ then*

1. $L_1(D_1, D_2) \leq \sum_i \max(\phi_i^1 - \phi_i^2, 0)$.

2. *If $A^* = \cup_{i:\phi_i^1 > \phi_i^2} C_i$ then $D_1(A^*) - D_2(A^*) = \sum_i \max(\phi_i^1 - \phi_i^2, 0)$.*

3. *If $\forall i,\ \phi_i^1 \neq \phi_i^2$ and $A$ is such that $D_1(A) - D_2(A) = L_1(D_1, D_2)$ then $\forall i,\ \theta_i(A \Delta A^*) = 0$.*

Figure 2: Demonstration of the DSC algorithm. Assume that $\Phi^1 = (0.4, 0.3, 0.3)$ for the orange, green and blue regions respectively and $\Phi^2 = (0.5, 0.1, 0.4)$. The green region maximizes the $L_1$ distance and therefore will be separated from the blue and orange. Conditioned on these two regions, the mixture coefficients are $\Phi^1_{\text{orange, blue}} = (4/7, 3/7)$ and $\Phi^2_{\text{orange, blue}} = (5/9, 4/9)$. The region that maximized this conditional $L_1$ is the orange regions that will be separated from the blue.

We conclude from Lemma 1 that if $D_1$ and $D_2$ were explicitly known and one could have found a classifier that best separates between the distributions, that classifier would not break clusters as long as the mixing coefficients

are not identical. In order for this to hold when the separation is applied recursively in the DSC algorithm it suffices to have that for every $I \subseteq [1, \ldots, K]$ if $|I| > 1$ and $i \in I$ then

$$\frac{\phi_i^1}{\sum_{i' \in I} \phi_{i'}^1} \neq \frac{\phi_i^2}{\sum_{i' \in I} \phi_{i'}^2}$$

to guarantee that at any stage of the algorithm clusters will not be split by the classifier (but may be sections of measure zero). This is also sufficient to guarantee that the leaves will contain single clusters.

In the case where data is provided through a finite sample then some book-keeping is needed. However, the analysis follows the same path. We show that with samples large enough, clusters are only minimally broken. For this to hold we require that the learning algorithm L separates the clusters according to this definition:

**Definition 1.** *For $I \subseteq [1, \ldots, K]$ let $c_I : X \mapsto \{\pm 1\}$ be such that $c_I(x) = 1$ if $x \in \cup_{i \in I} C_i$ and $c_I(x) = -1$ otherwise. A learning algorithm L separates $C_1, \ldots, C_K$ if for every $\epsilon, \delta > 0$ there exists $N$ such that for every $n > N$ and every measure $\nu$ over $X \times \{\pm 1\}$ with probability $1 - \delta$ over samples from $\nu^n$:*

    *1. The algorithm L returns an hypothesis $h : X \mapsto \{\pm 1\}$ and an error estimator $e \in [0, 1]$ such that $|\mathrm{Pr}_{x, y \sim \nu} [h(x) \neq y] - e| \leq \epsilon$*

    *2. h is such that*

$$\forall I, \quad \Pr_{x, y \sim \nu} [h(x) \neq y] < \Pr_{x, y \sim \nu} [c_I(x) \neq y] + \epsilon .$$

Before we introduce the main statement, we define what it means for a tree to cluster the mixture components:

**Definition 2.** *A* clustering tree *is a tree in which in each internal node is a classifier and the points that end in a certain leaf are considered a cluster. A clustering tree $\epsilon$-clusters the mixture coefficient $\theta_1, \ldots, \theta_K$ if for every $i \in 1, \ldots, K$ there exists a leaf in the tree such that the cluster $L \subseteq X$ associated with this leaf is such that $\theta_i(L) \geq 1 - \epsilon$ and $\theta_{i'}(L) < \epsilon$ for every $i' \neq i$.*

To be able to find a clustering tree, the two mixtures have to be different. The following definition captures the gap which is the amount of difference between the mixtures.

**Definition 3.** *Let $\Phi^1$ and $\Phi^2$ be two mixture vectors. The* gap, *$g$, between them is*

$$g = \min \left\{ \left| \frac{\phi_i^1}{\sum_{i' \in I} \phi_{i'}^1} - \frac{\phi_i^2}{\sum_{i' \in I} \phi_{i'}^2} \right| : I \subseteq [1, \ldots, K] \ and \ |I| > 1 \right\} .$$

*We say that $\Phi$ is b* bounded away *from zero if $b \leq \min_i \phi_i$.*

**Theorem 2.** *Assume that L separates $\theta_1, \ldots, \theta_K$, there is a gap $g > 0$ between $\Phi^1$ and $\Phi^2$ and both $\Phi^1$ and $\Phi^2$ are bounded away from zero by $b > 0$. For every $\epsilon^*, \delta^* > 0$ there exists $N = N(\epsilon^*, \delta^*, g, b, K)$ such that given two random samples of sizes $N < n_1, n_2$ from the two mixtures, with probability of at least $1 - \delta^*$ the DSC algorithm will return a clustering tree which $\epsilon^*$-clusters $\theta_1, \ldots, \theta_K$ when applied with the threshold $\tau = g/8$.*

## 4 Empirical evidence

We conducted several experiments with synthetic data to compare different methods when clustering in high dimensional spaces. The synthetic data was generated from three Gaussians with centers at points $(0, 0)$, $(3, 0)$ and $(-3, +3)$. On top of that, we added additional dimensions with normally distributed noise. In the first experiment we used unit variance for all dimensions. In the second experiment we skewed the distribution so that the variance in the other features is 5.

Two sets of mixing coefficients for the three Gaussians were chosen at random 100 times by selecting three uniform values from $[0, 1]$ and normalizing them to sum to 1. We generated

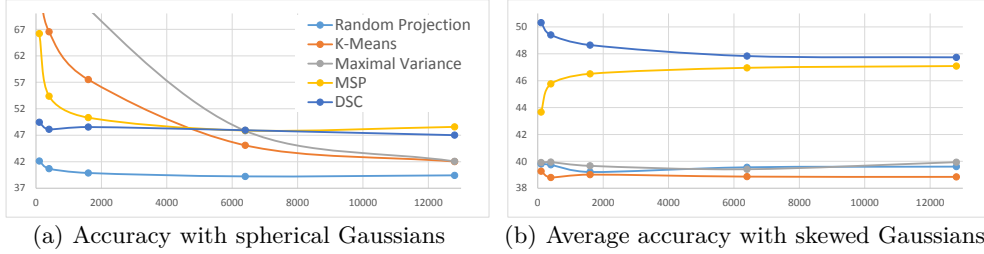

(a) Accuracy with spherical Gaussians     (b) Average accuracy with skewed Gaussians

Figure 3: **Comparison the different algorithms:** The dimension of the problem is presented in the X axis and the accuracy on the Y axis.

two samples with 80 examples each from the two mixing coefficients. The DSC and MSP algorithm received these two samples as inputs while the reference algorithms, which are not designed to use multiple samples, received the combined set of 160 points as input.

We ran 100 trials. In each trial, each of the algorithms finds 3 Gaussians. We then measure the percentage of the points associated with the true originating Gaussian after making the best assignment of the inferred centers to the true Gaussians.

We compared several algorithms. K-means was used on the data as a baseline. We compared three low dimensional projection algorithms. Following [5] we used random projections as the first of these. Second, following [11] we used PCA to project on the maximal variance subspace. MSP was used as the third projection algorithm. In all projection algorithm we first projected on a one dimensional space and then applied K-means to find the clusters. Finally, we used the DSC algorithm. The DSC algorithm uses the classregtree function in MATLAB as its learning oracle. Whenever K-means was applied, the MATLAB implementation of this procedure was used with 10 random initial starts.

Figure 3(a) shows the results of the first experiment with unit variance in the noise dimensions. In this setting, the Maximal Variance method is expected to work well since the first two dimensions have larger expected variance. Indeed we see that this is the case. However, when the number of dimensions is large, MSP and DSC outperform the other methods; this corresponds to the difficult regime of low signal to noise ratio. In 12800 dimensions, MSP outperforms Random Projections 90% of the time, Maximal Variance 80% of the time, and K-means 79% of the time. DSC outperforms Random Projections, Maximal Variance and K-means 84%, 69%, and 66% of the time respectively. Thus the p-value in all these experiments is $< 0.01$.

Figure 3(b) shows the results of the experiment in which the variance in the noise dimensions is higher which creates a more challanging problem. In this case, we see that all the reference methods suffer significantly, but the MSP and the DSC methods obtain similar results as in the previous setting. Both the MSP and the DSC algorithms win over Random Projections, Maximal Variance and K-means more than 78% of the time when the dimension is 400 and up. The p-value of these experiments is $< 1.6 \times 10^{-7}$.

## 5   Conclusions

The mixture problem examined here is closely related to the problem of clustering. Most clustering data can be viewed as points generated from multiple underlying distributions or generating functions, and clustering can be seen as the process of recovering the structure of or assignments to these distributions. We presented two algorithms for the mixture problem that can be viewed as clustering algorithms. The MSP algorithm uses multiple samples to find a low dimensional space to project the data to. The DSC algorithm builds a clustering tree assuming that the clusters are disjoint. We proved that these algorithms work under milder assumptions than currently known methods. The key message in this work is that when multiple samples are available, often it is best not to pool the data into one large sample, but that the structure in the different samples can be leveraged to improve clustering power.

## Footnotes

[1] the supremum is over all the measurable sets.

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
