[Reviews · NeurIPS 2013]

Submitted by Assigned_Reviewer_6

Mixture models (MM) assume that instances are drawn from a mixture of K component distributions with unknown coefficients. Topic models (TM), on the other hand, assume that samples/documents have different mixing weights of the underlying topic distribution over words. This paper tries to close the gap between MM and TM. Their proposed model assumes that several samples are drawn from the same underlying K distributions, but similar to TM, has different mixing weights and assume that instances are treated as feature vectors similar to MM.

This is a theory paper that provides two algorithms that can recover the underlying structure for this model. One algorithm, Multi Sample Projection (MSP), uses multiple samples to project high dimensional data to low dimensional space while keeping the means of the mixture components well separated. They utilize the assumption that, if we have sufficiently many mixtures, their means will span the affine space spanned by the K underlying means. The second algorithm, Double Sample Clustering (DSC), assumes disjoint clusters/supports but they do not make any assumption about the distributions. DSC is a tree-based approach such that each leaf represents a cluster and provided cluster “error” guarantees.

- This provides a nice contribution to clustering theory, linking MM and TM.
- They provide two new clustering algorithms that can learn from multiple mixtures.
- Their approach allows generalization to words not present in the training data. It would be nice to see experiments showing this important property.
- The experiments compared their approach with K-Means as baseline. They should also include topic modeling (e.g., LDA) as another baseline in the other extreme.
- The experiments only show when there are two sets of mixing coefficients. How does your algorithm fare with more than two sets?
- For the competing methods, why project them only to one dimensional space? Why not to K-1 dimensional space before applying K-means?

Comments to improve clarity of presentation:
- The use of the term “sample” in this paper was confusing. “Sample” in machine learning is commonly used to mean an instance. After re-reading the paper several times, I realized that the use of “sample” in this paper actually meant a population of instances with the same mixture weights. It would help to use a different term or at least define it early on to avoid confusion.
- In the introduction, make sure to indicate that the context of your citation to early work (“starting from [5]”) are works on clustering theory.
- Page 4, Line 191: “Let \mu is” to “Let \mu be”
- Page 5, Line 217: “assums”
Summary: This work provides two novel algorithms for finding multiple underlying distributions with different mixing component weights. They proved that these algorithms work under milder assumptions than competing methods. This provides a nice contribution to clustering theory, linking MM and TM.

Submitted by Assigned_Reviewer_7

This paper studies an interesting problem that lies between standard inference problems for mixture models, and for topic models. As a motivating example (borrowed from the paper), suppose that there are M hospitals, and we are interested in understanding various types of heart disease. Suppose that there are K types -- these represent the components in a standard mixture model. The main part of departure for this paper is, instead of grouping together all the samples into one set, make use of the fact that you know which patients are associated with which hospitals. In particular, we suppose that the patients in a hospital as a population are represented by a distribution over the K types of heart disease. This could be because certain types are more prevalent in particular geographic regions, etc.

Consider for example, the classic learning mixtures of Gaussians problem. There each sample is from one of the K components, but the trouble is that we do not know which component generated it. In this paper, the authors think of the samples as divided into M sets each one of which is represented by some intrinsic distribution on the K components. This is more similar in spirit to topic modeling (where it is crucial that we allow a document to be a distribution on multiple topics). Of course, this setting is also different than topic modeling since we are given a complete sample instead of a sparse vector of which words occur in a given document.

I think the question the authors study here is quite elegant, and I am not aware of any previous work. The authors give results somewhat analogous to the clustering-based approaches for mixtures of Gaussians, but with improved separation conditions. The first part of the paper is focused on recovering a low-dimensional subspace that roughly preserves the distance between the centers, and the second part assumes that the mixture components have disjoint supports. I think this second part of the paper is not so interesting, however. It is true that the separation conditions used in clustering mixtures of Gaussians mean that the components have almost disjoint supports, but it seems like only very strange distributions would have truly disjoint supports.
Summary: Overall, I think this paper is a nice contribution. The problem the authors study is interesting, and should be further studied. The results are novel, and have better separation conditions than existing work (which does not assume there are M types). However, the second part of the paper is not so natural in my opinion.

Submitted by Assigned_Reviewer_8

The problem of mixture model learning is considered, assuming the availability of samples from the same mixture components, but with different mixing weights. Two algorithms are described and analyzed.

The first, Multi Sample Projection (MSP), is a dimensionality reduction algorithm that seeks to find the subspace spanned by the means of the mixture components. MSP projects the data on the subspace spanned by the overall means of each of the available sets of samples. The theoretical analysis shows that the resulting procedure will maintain the separation between the components assuming the dimension isn't too large, and that there is sufficient variation in the mixing weight of each mixture component between the samples.

The second, Double Sample Clustering (DSC), assumes that exactly two sets of samples are available, that the mixture components have disjoint support, and that the (appropriately normalized) mixture weights for any subset of the components differs by some minimal amount between the two mixtures. The algorithm iteratively trains binary classifiers on reweighted subsets of the data sets, the intuition being that the support sets of the mixture components are regions where the sign of the difference of the densities of the two mixtures are constant.

The general idea of taking advantage of the "multiple sample" setting for clustering (there has to be a less ambiguous name for that by the way, but I can't think of one...) is interesting, and this paper may lead to further development of algorithms and theory on the topic.

The theoretical analysis of MSP and DSC are not much stronger than showing conditions for consistency, which is all that can be expected without further assumptions. As the authors point out, sharper rates could be proven for the preservation of distances in MSP if, for instance, a Gaussian mixture was assumed.

The numerical experiments provide encouraging evidence for the practical benefit of MSP and DSC in the multiple sample setting.

Small issues:

It is not clear to the reviewer why there is an exact equality in part 2 of Lemma 1, as opposed to a \leq.

Definition 1 is a bit confusing. My understanding is that here L refers to a base classifier that will be used in the iterations of the algorithm, but the conditions of the definition seem to be overly general (it allows any measure over X and +/-1, for one thing, which doesn't have anything to do with C_1,..., C_K; also condition 3 brings in C_I, even though, again, the input to the algorithm may have nothing to do with the clusters). Perhaps it would be clearer if Assumption 4 from section A.3 was used instead. Also, there is no mention of what delta is; I'm guessing there should be a "with probability 1-delta" somewhere, as well as an explanation that n is the number of samples to be used by the algorithm L (n was used for the number of dimensions earlier in the paper).
Summary: The idea of taking advantage of the "multiple sample" setting for clustering is interesting, and this paper may lead to further development of algorithms and theory on the topic. The two proposed algorithms are a good place to start, as evidenced by encouraging simulation results.
Author Feedback

Author rebuttal: We thank the reviewers for the thoughtful and encouraging reviews. The reviews include a number of good suggestions that we will try to incorporate in the final version of the paper. However, as noted by the reviewers, since this is a first attempt at a new problem, it is hard to address all aspects of it in a single work. Specifically, comparing to LDA is hard because LDA cannot be applied to the problem we experimented with since the distribution is continuous while LDA requires discrete distributions in the sense that the same word appears multiple times.